# SoftMatcha: A Soft and Fast Pattern Matcher for Billion-Scale Corpus Searches

**Hiroyuki Deguchi**[1,][*] **Go Kamoda**[2]**, Yusuke Matsushita**[3]**, Chihiro Taguchi**[4]**,**
**Kohei Suenaga**[3]**, Masaki Waga**[3]**, Sho Yokoi**[5,2,6]
[1]NAIST, [2]Tohoku University, [3]Kyoto University, [4]University of Notre Dame,
[5]NINJAL, [6]RIKEN
deguchi.hiroyuki.db0@is.naist.jp, go.kamoda@dc.tohoku.ac.jp,
ymat@fos.kuis.kyoto-u.ac.jp, ctaguchi@nd.edu
ksuenaga@kuis.kyoto-u.ac.jp, mwaga@fos.kuis.kyoto-u.ac.jp
yokoi@ninjal.ac.jp

## Abstract

Researchers and practitioners in natural language processing and computational linguistics frequently observe and analyze the real language usage in large-scale corpora. For that purpose, they often employ off-the-shelf pattern-matching tools, such as `grep`, and keyword-in-context concordancers, which is widely used in corpus linguistics for gathering examples. Nonetheless, these existing techniques rely on surface-level string matching, and thus they suffer from the major limitation of not being able to handle orthographic variations and paraphrasing—notable and common phenomena in any natural language. In addition, existing continuous approaches such as dense vector search tend to be overly coarse, often retrieving texts that are unrelated but share similar topics. Given these challenges, we propose a novel algorithm that achieves *soft* (or semantic) yet efficient pattern matching by relaxing a surface-level matching with word embeddings. Our algorithm is highly scalable with respect to the size of the corpus text utilizing inverted indexes. We have prepared an efficient implementation, and we provide an accessible web tool. Our experiments demonstrate that the proposed method (i) can execute searches on billion-scale corpora in less than a second, which is comparable in speed to surface-level string matching and dense vector search; (ii) can extract harmful instances that semantically match queries from a large set of English and Japanese Wikipedia articles; and (iii) can be effectively applied to corpus-linguistic analyses of Latin, a language with highly diverse inflections.

## 1 Introduction

Recent advances in natural language processing (NLP) and corpus linguistics are largely driven by the availability of massive text corpora (Gao et al., 2020; Biderman et al., 2023b; Raffel et al., 2019; Van Der Zwaan et al., 2017; Mcgillivray et al., 2020). The field of NLP, driven by the grand goal of building intelligent chatbots and machine translation systems, has achieved remarkable breakthroughs over the past decade, largely thanks to self-supervised representation learning from vast corpora (Mikolov et al., 2013; Pennington et al., 2014; Devlin et al., 2019a; Radford et al., 2019). Notable causal language models (LMs), capable of passing high-stakes exams (Katz et al., 2024; OpenAI, 2024; Jung et al., 2023), serving as general-purpose problem solvers (Brown et al., 2020), and engaging in realistic conversations with beloved characters (De Freitas, 2023; Chen et al., 2024), owe their foundational abilities to next-token prediction learned from large-scale data (Brown et al., 2020; Dubey et al., 2024; Riviere et al., 2024). In corpus linguistics (McEnery & Hardie, 2011), computational linguistics (Jurafsky & Martin, 2024), and digital humanities (Jensen, 2014)—disciplines aiming to uncover the scientific and computational principles of human language—such vast linguistic resources, consisting of the language use itself, have become more indispensable than ever in this era of large-scale corpora (Van Der Zwaan et al., 2017; Mcgillivray et al., 2020).

---

[*]Currently affiliated with NTT. E-mail: hiroyuki.deguchi@ntt.com

Table 1: *Hard* pattern matching (e.g., `grep`) and *soft* pattern matching (our method). Hard matching is based on *surface-level* comparison and does not match if, e.g., a synonym is used; Soft matching is based on *semantic* comparison and robust to such variations thanks to word embeddings.

| Pattern | **Theorem 1** | **John was born in** |
|---|---|---|
| *Hard* matching (e.g., `grep`) | $\cdots$ thanks to **Theorem 1** $\cdots$ | $\cdots$ **John was born in** 1345 $\cdots$ |
| *Soft* matching (**ours**) | $\cdots$ thanks to **Theorem 1** $\cdots$ | $\cdots$ **John was born in** 1345 $\cdots$ |
| | $\cdots$ **Theorem 3** holds because $\cdots$ | $\cdots$ **Edward had died in** May 1910 $\cdots$ |
| | $\cdots$ By **Lemma 5**, we may assume $\cdots$ | $\cdots$ **Robert was born in** England $\cdots$ |
| | $\cdots$ **Equation 1** describes $\cdots$ | $\cdots$ the Emperor **Henry, died in** 1125 $\cdots$ |

Given this context, the demand for efficient pattern matchers that enable rapid searches across massive corpora is higher than ever (Liu et al., 2024; Smadja, 1993). For example, when harmful information, misinformation, or memorized privacy information is generated by a large LM, it is necessary to identify the corresponding training instances where this information originated (Guo et al., 2022; Wang et al., 2024c; Ippolito et al., 2023; Biderman et al., 2023a). Another example is when researchers wish to determine which of two linguistic phenomena of interest is more frequent; conducting exhaustive searches across the largest available corpora is essential (Biber, 2015) for this purpose. Notably, many linguistic phenomena—word frequency as a simple example—follow a power-law distribution (Zipf, 1951; Heap, 1980; Kobayashi & Tanaka-Ishii, 2018). Consequently, only with vast corpora can we observe and analyze the various rare events residing in the long tail, which constitute the majority of linguistic phenomena.

One of the challenges when working with large corpora is that pattern-matching tools based on string matching (Hakak et al., 2019), such as `grep` (Bambenek & Klus, 2009) and `ripgrep` (Gallant, 2024), or linguist-oriented tools often referred to as KWIC (Anthony, 2013; Culy & Lyding, 2010; Schweinberger, 2024), primarily rely on surface-level exact matching as their core strategy. Because natural languages are characterized by their flexibility and richness in how humans can express similar concepts in different ways (McKeown, 1979; Witteveen & Andrews, 2019; Ganitkevitch et al., 2013), strictly exact string matching may not meet users' demands. For example, on top of the query word in standard spelling, it is often desirable to catch non-standard spellings as well, such as *how r u* instead of *how are you*, which are widespread particularly on the web and in texting (Schulz, 2018). Additionally, it is also desirable to catch different inflected word forms such as *sing*, *sang*, *sung*, *sings*, and *singing*, which differ only in their morphological features and share the same lemma (base form) (Don, 2014; Embick, 2015). Rule-based exact matching is particularly hard when the target language is morphologically complex and exhibits irregular inflectional patterns.

To resolve the mismatch between the symbolic nature of existing pattern matchers and the diverse orthographic and morphological variations inherent in natural language, we have developed a *soft* (or semantic) yet efficient pattern matcher (Section 3). The core strategy is based on the simple idea of *softening* the matching process from binary $\{0, 1\}$ values to continuous values, using word embeddings. By adopting inverted indexes and several other techniques, our tool can enumerate all softly matching instances in billion-scale corpora in less than a second. Table 1 shows specific examples of its operation. Our proposed method is capable not only of enumerating exact matches but also of flexibly listing semantically similar instances, even when their surface forms differ. For example, given the query "*Theorem 1*", the method can retrieve instances such as "*Lemma 5*". This characteristic substantially enhances key tasks in both NLP and corpus linguistics. For instance, it improves the filtering of harmful texts, and facilitates more efficient example retrieval, particularly for languages with complex morphological features (Section 4).

The contributions of our study are summarized as follows.

- We developed a *soft* (or semantic) pattern-matching algorithm—a relaxation of the exact string matching—using word embeddings (Section 3). By leveraging inverted indexing, we achieved high scalability (Sections 2.1, 2.3 and 3.3).[1]

---

[1] GitHub: https://github.com/softmatcha/softmatcha

- We developed an easy-to-use web demo to facilitate interaction with the soft matching algorithm.[2] This demo is particularly beneficial for NLP researchers and developers who want to analyze LMs in a data-driven manner, as well as for language learners and humanities researchers who are conducting language analysis on large corpora from the perspectives of corpus linguistics and digital humanities (Section 4.1)

- For quantitative evaluation, we verified that our method achieves complete enumeration in less than a second on billion-scale corpora, which are commonly used in training large LMs (Section 4.2). For instance, the running time for searching over an English corpus with 3.4B words was less than 0.1 seconds without GPUs. This performance is comparable in speed to dense vector search; moreover, our method enables the retrieval of examples closely aligned with specific queries, rather than just broad topical similarities.

- We conducted qualitative evaluations in specific scenarios to verify the usefulness of the proposed method in both the fields of NLP and corpus linguistics. In experiments assuming the training of LMs on large-scale corpora, we provided search examples for identifying and removing harmful instances contained within the corpus (Section 4.2). In experiments assuming the analysis of classical Western languages by linguists, we chose Latin—a language with highly complex inflections—as an example. We demonstrated that a corpus search with our tool can flexibly extract morphologically and semantically similar usage examples from the corpus (Section 4.3).

## 2 RELATED WORK

The procedures for finding sentences (or lines, documents) that match a given pattern (query) are prevalent across nearly all areas of computer science and data science, making it difficult to enumerate all related research. In this section, we review particularly relevant work in NLP and computational linguistics, which are the primary focus of this study, as well as in the closely related areas of string matching. Beyond the fields discussed here, we believe many other domains, such as information retrieval, time series analysis, and semantic web, also have connections to this research.

### 2.1 NATURAL LANGUAGE PROCESSING AND CORPUS SEARCH

The significant progress in NLP over the past decade is largely attributed to deep-learning–based self-supervised representation learning (Mikolov et al., 2013; Pennington et al., 2014; Bojanowski et al., 2017; Devlin et al., 2019a; Radford et al., 2019; Brown et al., 2020; Dubey et al., 2024). **The vast raw corpora** without label annotations serve as the source of such models' capabilities (Gao et al., 2020; Biderman et al., 2023b; Raffel et al., 2019), and, therefore, these corpora are continually referenced and analyzed for model improvement and evaluation (Biderman et al., 2023b). For instance, as LMs can memorize and elicit facts written in training corpora, they are reported to generate text containing privacy-related information (Huang et al., 2022; Li et al., 2023; Lukas et al., 2023) or content that could be used for terrorism or violence (Gehman et al., 2020; Schick et al., 2021; Kumar et al., 2023). In this context, Ippolito et al. (2023) work on filtering out verbatim memorization, requiring searches accross large-scale training corpora.[3] $N$**-gram substring pattern matchers**, as representative tools for corpus search in NLP, have been particularly well-developed even before the advent of deep learning (Sekine, 2008; Sekine & Dalwani, 2010). In terms of data structures, inverted indexes (Sekine & Dalwani, 2010; Rogozinski & Kuc, 2016), and suffix arrays including their variants (Sekine & Dalwani, 2010; Yamamoto & Church, 2001; Liu et al., 2024; Burrows et al., 1994; Ferragina & Manzini, 2005; Langmead et al., 2009) are typically employed. Our algorithm is based on inverted indexes due to its algorithmic requirements, which we discuss in Section 3. Here, regardless of which data structure is used, it is important to note that existing $n$-gram matching techniques assume exact surface-level matching, making it extremely challenging to search and enumerate all relevant sentences while handling the complex characteristics of natural language, such as paraphrasing, orthographic variation, and inflection. **Dense vector search** (Khattab & Zaharia, 2020; Karpukhin et al., 2020; Izacard et al., 2022; Wang et al., 2024a) has recently gained widespread popularity as the foundational technology for retrieval-augmented generation

---

[2] Demo: https://softmatcha.github.io. The screenshot of the demo is presented in Appendix C.

[3] Measuring influence of training corpus is also an important theme (Koh & Liang, 2017; Pruthi et al., 2020; Chen et al., 2021; Isonuma & Titov, 2024), but their application to vast corpora remains highly challenging.

(RAG) (Lewis et al., 2020; Khandelwal et al., 2020; Guu et al., 2020; Izacard et al., 2024). Dense vector search is highly compatible with approximate nearest neighbor search (Malkov & Yashunin, 2020; Jégou et al., 2011), and it offers a significant advantage in reducing the issue of hallucination (Ayala & Bechard, 2024; Gao et al., 2023) in scenarios requiring factual knowledge. However, dense vector search is a relatively "coarse" method, primarily used to retrieve documents that are topically similar, making it less suitable for the cases that require more specific $n$-gram level queries.

## 2.2 CORPUS LINGUISTICS AND CORPUS SEARCH

Corpus linguistics is a subfield of linguistics that utilizes corpora to study language based on examples of 'real-life' language use (McEnery & Wilson, 2001). Digital humanities, a closely related field, is an interdisciplinary domain that aims to advance humanities through the application of data and information science. In digital humanities as well, quantitative analysis of texts using corpora is also actively pursued (Jensen, 2014). **Corpora** consist of collections of texts or spoken language data (Van Der Zwaan et al., 2017; Mcgillivray et al., 2020), offering real-world examples of how language is used in various contexts. In most settings, an ideal corpus is large in size to ensure the statistical reliability of certain linguistic phenomena and to cover a broader range of language use, including rare occurrences such as low-frequency words (Ha et al., 2009; Coole et al., 2020). **Search tools** (Hockey & Martin, 1987; TEI Consortium, 2023) are an indispensable element of corpus linguistics because they enable the identification of linguistic patterns, such as word frequencies, collocations, and syntactic structures. Traditional search methods commonly used in corpus linguistics include exact matching and its extension with regular expressions (Jurafsky & Martin, 2024). However, given the nature of natural languages—with their morphological complexity and unlimited paraphrases and synonyms with varying surface forms—rule-based search methods face extreme difficulties in exhaustively enumerating instances that closely match a given query.

## 2.3 STRING MATCHING

From an algorithmic point of view, our method is closely related to *string matching* algorithms (Hakak et al., 2019). In what follows, we briefly give an algorithmic comparison with some of them and elucidate their relationship to ours. **Offline string matching** algorithms process the corpus text to build an index that accelerates future searches, a technique widely applied in search engines and information retrieval systems such as Elasticsearch (Rogozinski & Kuc, 2016) and Apache Lucene (Grand et al., 2020). Inverted indexing is a well-known approach in this domain (Zobel & Moffat, 2006), and our method serves as a *soft* generalization of a string matching algorithm based on inverted indexing. **Online string matching** algorithms do not rely on preprocessing and handle the corpus text on the fly. Efficient algorithms include the Knuth-Morris-Pratt, Karp-Rabin, and Boyer-Moore algorithms (Knuth et al., 1977; Karp & Rabin, 1987; Boyer & Moore, 1977). While widely used in tools such as `grep`, online string matching is also a basis of runtime verification (Bartocci et al., 2018), where system's execution data is monitored. Although it is possible to relax these algorithms with word embeddings, the number of *soft* word comparisons in such a relaxation would be linear in the size of the corpus. In contrast, our algorithm requires only constant *soft* word comparisons, thanks to indexing. See Section 3.3 for the complexity analysis. **Approximate string matching** allows flexible matching, typically using edit distance to accommodate noisy or incomplete data (Navarro, 2001). Tools like `agrep` (Wu & Manber, 1992b;a) perform online matching that tolerates mismatches. Indexing-based algorithms are also proposed for approximate string matching (Boytsov, 2011). As a relaxation of exact string matching, approximate string matching is orthogonal to ours: approximate string matching focuses on relaxing surface-level comparison, while our approach focuses on semantic-level similarity based on word embeddings. For example, morphologically irregular forms such as "person" and "people", only differing in plurality, will likely be missed in approximate string matching, while our methods are able to catch them.

## 3 OUR ALGORITHM FOR SOFT PATTERN MATCHING

### 3.1 OVERVIEW

**Key aspect of design: Hard vs. Soft.** One key aspect we considered in designing the algorithm for soft pattern matching is *hard computation vs. soft computation*. *Hard* (or exact) pattern matching

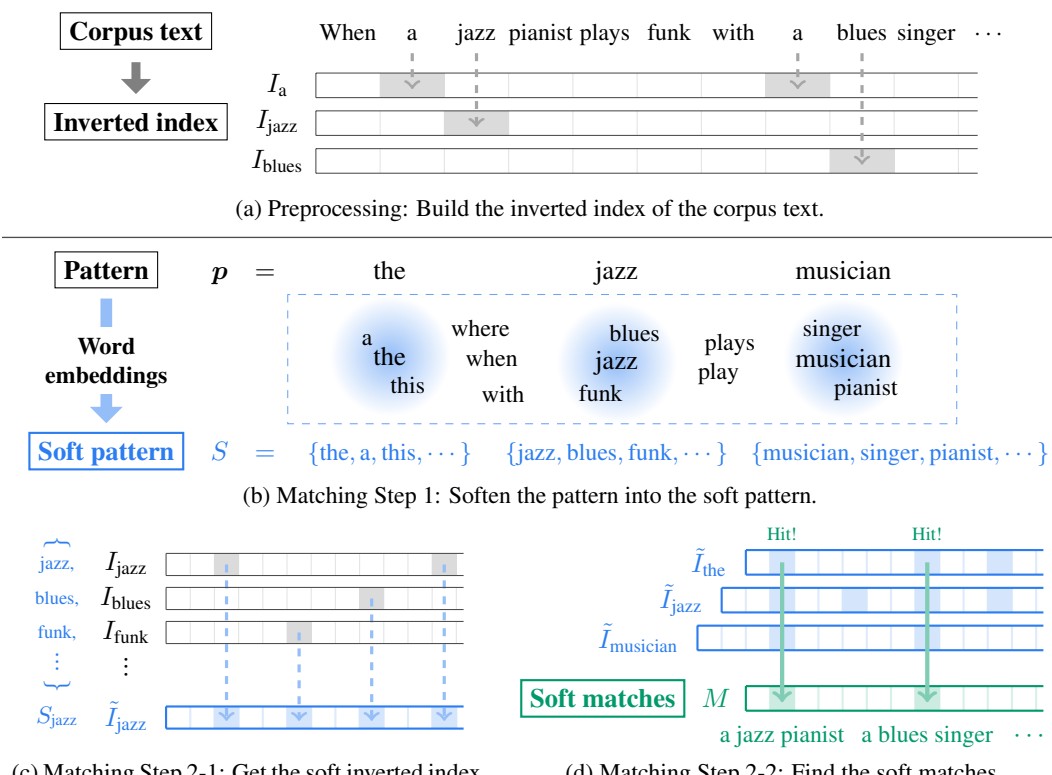

(a) Preprocessing: Build the inverted index of the corpus text.

(b) Matching Step 1: Soften the pattern into the soft pattern.

(c) Matching Step 2-1: Get the soft inverted index.

(d) Matching Step 2-2: Find the soft matches.

Figure 1: Illustration of our algorithm for soft pattern matching.

(such as `grep`) can process large texts blazingly fast. This is because hard matching requires only "*hard* computation", e.g., bitwise comparison, which is highly efficient. *Soft* pattern matching, on the other hand, involves "*soft* computation", e.g., cosine similarity of two vectors. Since such *soft* computation involves many floating-point arithmetic operations over high-dimensional vectors, it is typically much slower than *hard* computation. For instance, just taking the cosine similarity of a pair of word embeddings can be as expensive as thousands of simple bitwise operations.

**Overview of our algorithm.** With this in mind, we designed the algorithm for soft pattern matching, which is precise and efficient. Our algorithm is illustrated in Figure 1. Our key idea is to run soft computation *only over the vocabulary*, not over the whole text as a naive algorithm would do. More specifically, our algorithm works in the following two steps: For each query, (Step 1) it first performs *soft* computation over the *vocabulary*, *softening* the pattern into the *soft pattern* $S_1 S_2 \cdots S_n$ (Fig. 1b); (Step 2) then it performs *hard* computation that finds the *exact* positions in the corpus text that match the soft pattern (Figs. 1c and 1d). Here, the soft pattern $S_1 S_2 \cdots S_n$ is the key data in play. It consists of the set of vocabulary words that *softly* match each word of the pattern in terms of word embeddings. The amount of soft computation (Step 1) is small, given that the vocabulary size is typically much smaller than the size of the corpus text. The amount of hard computation (Step 2) is also quite small, because it only handles filtered positions, not the whole corpus text.

## 3.2 DETAILS

**Problem formulation.** First, we formulate the *soft* pattern-matching problem by simply relaxing the classical *hard* (or exact) pattern-matching problem (Hakak et al., 2019) as follows:

---

**Algorithm 1:** Our soft pattern-matching algorithm.

---

**Given** : The corpus text $\boldsymbol{t} = t_1 t_2 \cdots t_N$ and its inverted index $I_v \subseteq \{1, \ldots, N\}$ over each vocabulary word $v \in \mathcal{V}$ such that $i \in I_v$ if and only if $v = t_i$.

**Input** : The phrase pattern $\boldsymbol{p} = p_1 p_2 \cdots p_n$ and the threshold $\alpha \in (0, 1]$.

**Output :** The set of soft match positions $M \subseteq \{1, \ldots, N\}$, such that $i \in M$ holds if and only if $p_k \approx_\alpha t_{i+k-1}$ holds over any $k = 1, \ldots, n$.

```
// Step 1: Soften the pattern p₁p₂···pₙ into the soft pattern S₁S₂···Sₙ
```
1 **for** $k \leftarrow 1 \ldots n$ **do**
2     $S_k \leftarrow \varnothing$
3     **for** $v \in \mathcal{V}$ **do**
4        **if** $v \approx_\alpha p_k$ **then** $S_k \leftarrow S_k \cup \{v\}$
```
// Step 2-1: Get the soft inverted index Ĩₖ by relaxing I with Sₖ
```
5 **for** $k \leftarrow 1 \ldots n$ **do**
6     $\tilde{I}_k \leftarrow \varnothing$
7     **for** $v \in S_k$ **do** $\tilde{I}_k \leftarrow \tilde{I}_k \cup I_v$
```
// Step 2-2: Get the complete matching M by aggregating Ĩₖ
```
8 $M \leftarrow \tilde{I}_1$
9 **for** $k \leftarrow 2 \ldots n$ **do** $M \leftarrow M \cap \big\{ i - (k-1) \,\big|\, i \in \tilde{I}_k \big\}$
10 **return** $M$

---

> **The soft pattern-matching problem.**
> INPUT: The corpus text $\boldsymbol{t} = t_1 t_2 \cdots t_N \in \mathcal{V}^*$, the pattern $\boldsymbol{p} = p_1 p_2 \cdots p_n \in \mathcal{V}^*$, and the threshold $\alpha \in (0, 1]$
> OUTPUT: The set of soft match positions $M$ with respect to the soft equivalence $\approx_\alpha$, i.e., the set $M = \big\{ i \in \{1, \ldots, N\} \,\big|\, \forall k \in \{1, \ldots, n\}. \, p_k \approx_\alpha t_{i+k-1} \big\}$

We define here the *soft equivalence* $v \approx_\alpha v'$ between words as $v \approx_\alpha v' \overset{\triangle}{\iff} \cos(E(v), E(v')) \geq \alpha$, to capture the similarity in terms of *word embeddings* $E$. Here, we let $\mathcal{V} = \{v_1, v_2, \ldots, v_L\}$ be the vocabulary and write $E(v) \in \mathbb{R}^D$ for the word embedding of a word $v \in \mathcal{V}$. Also, we write $\cos(\boldsymbol{e}, \boldsymbol{e}')$ for the cosine similarity $\cos(\boldsymbol{e}, \boldsymbol{e}') \triangleq \frac{\boldsymbol{e} \cdot \boldsymbol{e}'}{\|\boldsymbol{e}\| \|\boldsymbol{e}'\|}$ of word embeddings $\boldsymbol{e}, \boldsymbol{e}' \in \mathbb{R}^D$.

**Preprocessing.** Before processing queries, our algorithm preprocesses the whole corpus text to compute the *inverted index* $I$ (Fig. 1a), following a standard technique in search engine indexing (Zobel & Moffat, 2006). The inverted index $I$ maps each vocabulary word $v \in \mathcal{V}$ to the set of positions $I_v \subseteq \{1, \ldots, N\}$ that represents the occurences of the word $v$ in the corpus text $\boldsymbol{t}$, that is, the set $I_v = \big\{ i \in \{1, \ldots, N\} \,\big|\, v = t_i \big\}$.

**Our algorithm.** Algorithm 1 and Figures 1b to 1d outline our algorithm for soft pattern matching. Our algorithm works in the following two steps:

1. For each query, it performs *soft* computation over the *vocabulary*, *softening* the pattern $p_1 p_2 \cdots p_n$ into the *soft pattern* $S_1 S_2 \cdots S_n$ (lines 1 to 4 in Algorithm 1, Figure 1b). Here, $S_k$ is the set of vocabulary words that *softly* matches each word $p_k$ of the pattern in terms of the *soft* equivalence $\approx_\alpha$, i.e., $S_k = \big\{ v \in \mathcal{V} \,\big|\, v \approx_\alpha p_k \big\}$.

2. Then, it performs *hard* computation that computes the *exact* set of positions $M$ in the corpus text that matches the soft pattern $S_1 S_2 \cdots S_n$ by the following two sub-steps:

   2-1. Compute the *soft inverted index* $\tilde{I}$, which maps each pattern word $p_k$ to the union $\tilde{I}_k = \bigcup_{v \in S_k} I_v$ of the exact inverted index $I$ over $S_k$ (lines 5 to 7, Fig. 1c). The set $\tilde{I}_k$ agrees with the set of positions that softly match the pattern word $p_k$, i.e., $\tilde{I}_k = \big\{ i \in \{1, \ldots, N\} \,\big|\, t_i \approx_\alpha p_k \big\}$.

   2-2. Output the set of *soft matches* $M$ by computing the shifting intersection $M = \bigcap_{k=1}^n \big\{ i - (k-1) \,\big|\, i \in \tilde{I}_k \big\}$ over the soft inverted index $\tilde{I}$ (lines 8 to 9,

Fig. 1d). The resulting set precisely agrees with the expected set, i.e., $M = \{ i \in \{1, \ldots, N\} \mid \forall k \in \{1, \ldots, n\}. p_k \approx_\alpha t_{i+k-1} \}$.

We perform soft pattern matching using the index $I$. First, for each word $p_k$ in $\boldsymbol{p}$, we construct the set $\tilde{I}_k$ indicating the positions of the words in $\boldsymbol{t}$ matching $p_k$ (lines 1 to 4): we compare each word $v \in \mathcal{V}$ in the vocabulary with $p_k$ (line 4); we add $I_v$ to $\tilde{I}_k$ if we have $v \approx_\alpha p_k$ (line 7). Then, we construct the result $M$ by aggregating each $\tilde{I}_k$ considering the position $k$ of $p_k$ in $\boldsymbol{p}$.

**Inverted index vs. Suffix array.** Our algorithm uses an inverted index rather than a suffix array (Yamamoto & Church, 2001), which is crucial. A suffix array manages *exact sequences* of characters (or tokens) in the corpus text, for which *softening* patterns cannot be performed efficiently. In contrast, entries of an inverted index $I_v$ just have the occurrences of *each* word type $v$ and can be relaxed for soft matching simply by merging the sets, as illustrated in Fig. 1c.

### 3.3 FORMAL ANALYSIS

How efficient is our algorithm? To answer this with theoretical guarantees, we analyze the time and space complexity of the algorithm. Overall, the high-level observation is that the corpus text size $N$ affects the time and space required for preprocessing and soft matching only *linearly*.

**Preprocessing.** The time complexity for constructing the inverted index $I$ is $\mathcal{O}(L \times N)$, consisting of $L \times N$ exact word comparisons, since each vocabulary word $v \in \mathcal{V}$ is compared with each word $t_i$ in the corpus text $\boldsymbol{t}$. Notably, the constant factor here is typically very low, as only bitwise comparison is required. The total space required to store the inverted index $I$ is only $\mathcal{O}(N + L)$, by simply storing the list of positions (of space $\mathcal{O}(|I_v|)$) for each vocabulary word $v \in \mathcal{V}$. This is because each position $i \in \{1, 2, \ldots, N\}$ in the text $\boldsymbol{t}$ occurs exactly once in $I$ and hence $N = \sum_{v \in \mathcal{V}} |I_v|$. The space complexity is also $\mathcal{O}(N + L)$, since no extra space is required.

**Soft matching.** Step 1 for softening the pattern takes $\mathcal{O}(n \times L)$ in time, where the dominant part is $n \times L$ soft word comparisons (taking the cosine similarity of word embeddings) between each pattern word $p_k$ and each vocabulary word $v \in \mathcal{V}$ (line 4). The space complexity is $\mathcal{O}(\sum_{k=1}^{n} |S_k|)$. Notably, the time and the space for Step 1 are independent of the corpus text size $N$. Step 2 for finding the set of soft matches takes $\mathcal{O}(K)$ in time and space, where $K$ is the total size of the soft inverted index $K \triangleq \sum_{k=1}^{n} |\tilde{I}_k|$ (or roughly the total number of 'candidate' positions for matches). Remarkably, this is only linear to the corpus text size $N$.

## 4 EMPIRICAL EVALUATION

We address the following research questions in the experiments:

**RQ1:** Does `SoftMatcha` scale to a billion-scale corpus, which is the typical size of training corpus for large causal LMs? (Section 4.2)

**RQ2:** Does `SoftMatcha` perform as expected in typical scenarios in NLP and corpus linguistics? (Sections 4.2 and 4.3)

### 4.1 IMPLEMENTATION

We implemented the algorithm in Section 3 as a tool named `SoftMatcha`.[4] Our implementation adopts the following designs for efficiency. Firstly, we design our inverted index using a sparse matrix in a compressed sparse row (CSR) format to reduce memory consumption and enable efficient access to the index for each word, i.e., $I_v$. The index $I$ is represented by a one-dimensional array, with the positions of each word contiguously allocated in memory. Secondly, we compile some time-consuming operations to native code using NUMBA (Lam et al., 2015). Specifically, line 4 in Algorithm 1 calculates word embedding similarities over $n \times L$ times, i.e., the time complexity is

---

[4] The word *Matcha* means powdered green tea in Japanese and is here a pun for 'matcher'. *Matcha softo* in Japanese means soft-serve ice cream of green tea flavor.

$\mathcal{O}(n \times L \times D)$, where $D$ is the dimension of each word embedding. To speed up this calculation, we leverage the vectorized instruction set (SIMD) for parallel processing. In addition, finding the intersection of two large sets in line 9 in Algorithm 1 is computationally expensive. We employ just-in-time (JIT) compilation and parallel loops for efficient comparisons of elements. To ensure reproducibility, we release our source code on GitHub (Footnote 1) and the package on PyPI[5].

We provide a demo environment (Footnote 2) for users to explore the capabilities of our method and experience how diverse and linguistically natural the search results are and how quickly results can be obtained. The corpora used in experiments in Sections 4.2 and 4.3 are available. To prevent excessive output, for English and Japanese, only subsets of the corpora have been incorporated. Furthermore, search results are presented in smaller batches—to improve usability—with additional results available upon request, rather than all at once.

## 4.2 CASE STUDY IN NATURAL LANGUAGE PROCESSING — BILLION-SCALE CORPUS SEARCH

In this section, we simulate scenarios where NLP researchers use `SoftMatcha` on billion-scale English and Japanese corpora. We focus on two types of evaluations: (i) quantitative evaluation — we assess the running time using large data comparable in size to those used for training standard large LMs; and (ii) qualitative evaluation — we examine the tool's effectiveness in detecting toxic instances within large LM training corpora.

**Why English and Japanese?** English is the dominant language in modern NLP and accounts for the largest portion of training data for large LMs (Brown et al., 2020; Chowdhery et al., 2022; Touvron et al., 2023). In contrast, Japanese presents a typologically distinct language, with orthographic and structural features that differ significantly from English, such as: (i) character types (alphabetic vs. hiragana, katakana, and kanji); (ii) syntax (SVO vs. SOV, head-initial vs. head-final); and (iii) word segmentation (space-separated vs. no word separation) (Haspelmath et al., 2005). Furthermore, the substantial size of Japanese corpora (Wikipedia, 2024) makes it suitable for testing the scalability of our approach.

**Setup. Corpora:** we utilized the LLM-jp corpus v2.0 (LLM-jp et al., 2024), which contains organized collections of Wikipedia articles in both English (3.4B words) and Japanese (1.1B words). **Word embeddings:** we used GloVe `glove-wiki-gigaword-300` (Pennington et al., 2014) with a threshold $\alpha = 0.55$ for the English word embeddings, and fastText `facebook/fasttext-ja-vectors` (Grave et al., 2018) with a threshold $\alpha = 0.50$ for the Japanese word embeddings. **Baseline methods:** we compared the search results of `SoftMatcha` with those of exact matching, i.e., pattern matching with a standard inverted index, and dense vector search using `intfloat/multilingual-e5-large` model (Wang et al., 2024b). In dense vector search, we encoded each training instance[6] in the corpus and built the graph-based index using hierarchical navigable small worlds (HNSW) (Malkov & Yashunin, 2020) with the FAISS implementation (Douze et al., 2024) for the approximate nearest neighbor search. In HNSW, the number of edges was 32, the efConstruction was set to 40, and efSearch was configured as 16 (Malkov & Yashunin, 2020). We prepended the prefix "passage: " to each instance and "query: " to each query, following Wang et al. (2024b), and truncated any instances that exceeded 512 tokens. The text embeddings were calculated by averaging the contextualized token embeddings. **Computational environment:** we measured the running time on 152 core CPUs (Intel® Xeon® Platinum 8368 CPU @ 2.40GHz) and a 226 GiB main memory for the exact matching and `SoftMatcha`, and on 8 NVIDIA A100 GPUs for the dense vector search.

**Running time.** We measured the indexing time and search time in both the English and Japanese Wikipedia articles. Table 2 demonstrates that `SoftMatcha` is faster than dense vector search in both indexing and search time and takes the same indexing time as exact matching. Next, we investigated the relationship between the search speed and the corpus size. We constructed subsets of the English Wikipedia, whose sizes are $\{10^{-3}, 10^{-2}, 10^{-1}\}$ times that of the original corpus of 3.4B tokens, by randomly sampling from the original corpus, and measured the search time with each subset. Figure 2 shows the results. We observed that the search time increased only sublinearly,

---

[5] PyPI: https://pypi.org/project/softmatcha
[6] A training instance of LLM is a chunk of texts, which may consist of multiple sentences.

Table 2: Running time (sec) of indexing and search in the English and Japanese Wikipedia articles.

|  | En (3.4B words) | | Ja (1.1B words) | |
|---|---|---|---|---|
|  | Indexing | Search | Indexing | Search |
| Exact matching | 685.8 | 0.005 | 242.5 | 0.022 |
| SoftMatcha | 685.8 | 0.098 | 242.5 | 0.055 |
| Dense vector search | 1036.5 | 0.389 | 320.4 | 0.283 |

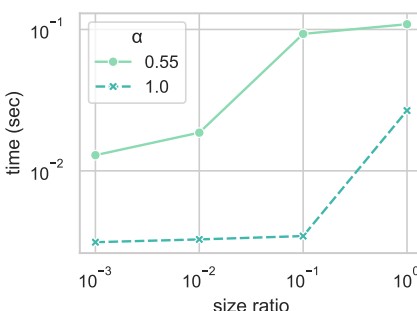

Figure 2: Running time on subsampled English Wikipedia corpora.

Table 3: Results of billion-scale corpus search in the English and Japanese Wikipedia articles.

|  | Exact matching | SoftMatcha | Dense vector search |
|---|---|---|---|
| Query: "*homemade bombs*" (2 tokens) in English Wikipedia (3.4B words) | | | |
| # Hits | 107 | 1,473 | n/a (depends on the top-$k$) |
| Match examples | *homemade bombs* | *homemade bombs* | Article: Survival Under Atomic Attack |
|  |  | *home-made grenades* | Article: Mark 24 nuclear bomb |
|  |  | *homemade missiles* | Article: List of common misconceptions |
| Query: "手製爆弾 (*homemade bombs*)" (2 tokens) in Japanese Wikipedia (1.1B words) | | | |
| # Hits | 27 | 42 | n/a (depends on the top-$k$) |
| Match examples | 手製爆弾 (*homemade bombs*) | 手製爆弾 (*homemade bombs*) | Article: 花火 (*fireworks*) |
|  |  | 手製手榴弾 (*home-made grenades*) | Article: 手持ち花火 (*consumer fireworks*) |

not linearly, with respect to the increase in corpus size. We thus confirmed that our algorithm works effectively for billion-scale corpus searches.

**Search results.** Table 3 shows the results of the billion-scale corpus search. We queried "*homemade bombs*" in the English Wikipedia corpus and "手製爆弾 (*homemade bombs*)" in the Japanese Wikipedia corpus. In the dense vector search, we selected some retrieved examples from the top-10 search results. The table demonstrates that our SoftMatcha extends the exact matching. Note that the results of exact matching are a subset of the results of SoftMatcha. In Japanese, the dense vector search retrieved an article of "花火" (*fireworks*), which does not contain the contents related to the query. In contrast, SoftMatcha lists all contents that have the query pattern or its similar pattern. In addition, while dense vector search allows semantic similarity search, it cannot identify where the query pattern is located in a text. To summarize, a text that exactly contains the query pattern is not always retrieved in dense vector search, i.e., the recall is not always 100%, while exact matching and SoftMatcha can return all texts that contain the query pattern, and SoftMatcha also enables matching of semantically similar patterns.

### 4.3 CASE STUDY IN CORPUS LINGUISTICS — RETRIEVING LATIN EXAMPLES

**Why Latin?** Latin is a morphologically complex fusional language where a lemma (dictionary form) may exhibit numerous different word forms depending on the morphological features (*e.g.*, voice, mood, tense, aspect, person, and number for verbals; gender, number, and case for nominals) that the word bears. For example, a transitive verb may conjugate to more than 100 different finite verb forms. This morphological complexity makes it harder to search through a corpus by exact matching. Furthermore, due to Latin's philological significance and the vast body of accumulated literature, there has been a persistent demand for advanced search tools to facilitate corpus analysis (Bamman & Smith, 2012). For these reasons, Latin is a suitable touchstone to test the utility of our proposed soft pattern matcher, particularly for humanities researchers and language learners.

Table 4: Latin match examples by `SoftMatcha` with queries *factus est* 'he/it was done/finished/made' and *equus est* 'it is a horse'. The top row of an interlinear gloss represents words (and the cosine similarity between the matched word and its corresponding query word), the middle row their morphological analysis, and the bottom row the free translation. Morphological matches are highlighted in pink, semantics matches in blue, and exact matches in green.

| Query | | | Query | |
|---|---|---|---|---|
| *factus* | *est* | | *equus* | *est* |
| do.PASS.PF.PTCP-M.NOM.SG | be.IND.PRS.3SG | | horse.M.NOM.SG | be.IND.PRS.3SG |
| 'he/it is done/finished/made' | | | 'it is a horse' | |
| **Matches** | | | **Matches** | |
| 0.56 | 0.56 | | 0.48 | 1.00 |
| *facta* | *sunt* | | *bos* | *est* |
| do.PASS.PF.PTCP-N.NOM.PL | be.IND.PRS.3PL | | cow.F.NOM.SG | be.IND.PRS.3SG |
| 'they are done' or 'they are facts' | | | 'it is a cow.' | |
| 0.53 | 0.58 | | 0.44 | 0.63 |
| *mortuus* | *esset* | | *currus* | *fuit* |
| die.ACT.PF.PTCP-M.NOM.SG | be.SUB.IMPF.3SG | | chariot.M.NOM.SG | be.IND.PF.3SG |
| 'he/it was dead' | | | 'it was a chariot' | |
| 0.65 | 0.65 | | 0.49 | 0.58 |
| *creatus* | *erat* | | *Minotaurus* | *esset* |
| create.PASS.PF.PTCP-M.NOM.SG | be.IND.IMPF.3SG | | Minotaur.M.NOM.SG | be.SUB.IMPF.3SG |
| 'he/it was created' | | | 'it would be the Minotaur' | |

**Setup. Corpora:** we used two corpora: one from the Perseus Project (Crane, 2023) (5M tokens) and the Augustinian Sermon Parallelism (ASP) Dataset (Bothwell et al., 2023) (0.1M tokens). **Word embeddings:** we used the pre-trained fastText embeddings `facebook/fasttext-la-vectors` (Grave et al., 2018).

**Search results.** Table 4 shows the returned matches with the queries *factus est* ('it/he is done' or 'it/he is made') and *equus est* ('it is a horse'). It is evident that `SoftMatcha` effectively links the queries to semantically similar words, as seen in *mortuus* 'dead' and *creatus* 'created' matched with the query *factus* 'done, finished, made', and *bos* 'cow', *currus* 'chariot', and *Minotaurus* 'Minotaur' with *equus* 'horse'. Interestingly, the tool is also able to catch different word forms with different morphological features while sharing the same lemma, which are highlighted in pink. For example, although the Latin copula verb in the query *est* 'is' exhibits highly irregular conjugation patterns, the matches successfully include their inflected forms such as *sunt*, *esset*, *erat*, and *fuit*. Furthermore, the matches *mortuus* and *creatus* are not only semantically similar to the query word *factus* but also have morphological features in common (perfect, participle, masculine, nominative, and singular).

## 5 CONCLUSION

In this paper, we propose a new pattern-matching algorithm that can flexibly handle the orthographic diversity of natural languages while also performing efficiently on large-scale corpora. Our algorithm, combining word embeddings and inverted indexing, achieves inference speed independent of corpus size. We have also developed and released a simple-to-use web demo for researchers and practitioners. For quantitative evaluation, we confirm that the developed tool can enumerate all search results within one second on billion-scale corpora, a typical scenario in training large LMs. For qualitative evaluation, we assess the tool in typical scenarios in NLP (detecting harmful examples from large-scale Japanese and English Wikipedia corpora) and computational linguistics (example retrieval from Latin, a language with highly diverse inflections), observing that our method can retrieve semantically similar examples that hard pattern matchers would miss.

ACKNOWLEDGMENTS

We conducted experiments using the language and computational resources of LLM-jp (`https://llm-jp.nii.ac.jp`). This research was supported in part by JSPS KAKENHI JP24KJ0133, JSPS KAKENHI JP23K24910, JST ACT-X JPMJAX200S, JST ACT-X JPMJAX200U, JST FOREST JPMJFR2331, JST PREST JPMJPR22CA, and JST CREST JPMJCR2012.

ETHICS STATEMENT

The corpora used for training large LMs, composed of web corpora, including Common Crawl (Common Crawl, 2024) and digitized book data such as (Zhu et al., 2015), can be said to encompass a substantial portion of humanity's linguistic knowledge. Inevitably, these corpora include a significant amount of ethically problematic content. This includes personal information (Subramani et al., 2023), as well as information that could be "beneficial" for violence and terrorism (or in more conventional terms, harmful information) (Albalak et al., 2024). Moreover, as these corpora continue to grow exponentially and our languages possess numerous paraphrases, comprehensively identifying such harmful learning resources is far from a simple task. Our research aims to substantially simplify this crucial activity for human ethics: identifying harmful learning instances within vast linguistic resources Section 4.2. We hope that, by leveraging our tool, NLP researchers and developers will contribute to providing LLMs as a safe and reliable social infrastructure.

REPRODUCIBILITY STATEMENT

Our source code is available on GitHub (Footnote 1), and we released an installable package on PyPI (Footnote 5). In all experiments, we used open datasets. Further details on the embeddings, corpora, computational environment, and protocols used in experiments are explained in Section 4.

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

## A    OTHER EXAMPLES OF SOFT MATCHES IN LATIN

Table 5 shows several additional examples of Latin soft matches by `SoftMatcha`.

## B    LIST OF GLOSSING ABBREVIATIONS

Table 6 lists the glossing abbreviations used in this paper.

## C    WEB INTERFACE

Figures 3 to 5 shows screenshots of our demo tool.

Table 5: Additional examples of soft matches in Latin and their scores returned by `SoftMatcha`. The top row of an interlinear gloss represents words (and the cosine similarity between the word and its corresponding query word), the middle row their morphological analysis, and the bottom row the free translation. Morphological matches are highlighted in pink, semantic matches in blue, and exact matches in green.

| Query | |
|---|---|
| ***non*** | ***potest*** |
| not | can.IND.PRS.3SG |
| '*he/she/it cannot*' | |

| Matches | |
|---|---|
| 1.00 | 0.59 |
| ***non*** | ***possit*** |
| not | can.SUB.PRS.3SG |
| '*he/she/it cannot*' | |
| 0.53 | 0.52 |
| ***nec*** | ***posse*** |
| nor | can.INF.PRS |
| '*not to be able*' | |

| Query | |
|---|---|
| ***homo*** | ***sapiens*** |
| human.M.NOM.SG | wise.M.NOM.SG |
| '*a wise human*' | |

| Matches | |
|---|---|
| 1.00 | 0.39 |
| ***homo*** | ***honestus*** |
| human.M.NOM.SG | noble-M.NOM.SG |
| '*an honorable human*' | |
| 0.47 | 0.40 |
| ***vir*** | ***sincerus*** |
| man.M.NOM.SG | pure-M.NOM.SG |
| '*a pure man*' | |

| Query[7] | |
|---|---|
| ***post*** | ***meridiem*** |
| after | noon.M.ACC.SG |
| '*after noon*' | |

| Matches | |
|---|---|
| 0.68 | 1.00 |
| ***ante*** | ***meridiem*** |
| before | noon.M.ACC.SG |
| '*before noon*' | |
| 0.51 | 0.52 |
| ***contra*** | ***septentrionem*** |
| against | Ursa.Major.F.ACC.SG |
| '*against Ursa Major*', i.e., '*facing north*' | |

| Query[8] | |
|---|---|
| ***bellum*** | ***gallicum*** |
| war.N-NOM.SG | gallic-N.NOM.SG |
| '*the Gallic war*' | |

| Matches | |
|---|---|
| 1.00 | 0.44 |
| ***bellum*** | ***etruscum*** |
| war.N-NOM.SG | Etruscan-N.NOM.SG |
| '*the Etruscan war*' | |
| 0.49 | 0.45 |
| ***contra*** | ***Caesarem*** |
| against | Caesar.M-ACC.SG |
| '*against Caesar*' | |

| Query[9] | |
|---|---|
| ***quo*** | ***vadis*** |
| where | go.IND.PRS.2SG |
| '*where do you go*' | |

| Matches | |
|---|---|
| 0.42 | 1.00 |
| ***ibi*** | ***vadis*** |
| there | go.IND.PRS.2SG |
| '*you go there*' | |
| 0.50 | 0.48 |
| ***autem*** | ***vocaris*** |
| but | summon-PASS.IND.PRS.2SG |
| '*but you are summoned*' | |

| Query[10] | | |
|---|---|---|
| ***quod*** | ***erat*** | ***demonstrandum*** |
| which | be.IND.IMPF.3SG | show.GER-N.NOM.SG |
| '*which was to be shown*' | | |

| Matches | | |
|---|---|---|
| 0.46 | 1.00 | 0.41 |
| ***haec*** | ***erat*** | ***forma*** |
| this.F.NOM.SG | be.IND.IMPF.3SG | form.F.NOM.SG |
| '*this was the form*' | | |
| 1.00 | 0.54 | 0.47 |
| ***quod*** | ***postea*** | ***accipiamus*** |
| which | afterwards | accept.ACT.SUB.PRS.1PL |
| '*which we shall accept later*' | | |

Table 6: List of the gloss abbreviations used in this paper.

| Gloss | Meaning |
|---|---|
| 1, 2, 3 | 1st, 2nd, 3rd person, respectively |
| ACC | accusative |
| ACT | active voice |
| F | feminine (gender) |
| FUT | future |
| GER | gerundive |
| IMPF | imperfective |
| IND | indicative mood |
| M | masculinie (gender) |
| N | neuter (gender) |
| NOM | nominative case |
| PASS | passive voice |
| PF | perfective |
| PL | plural |
| PRS | present tense |
| PTCP | participle |
| SG | singular |
| SUB | subjunctive mood |

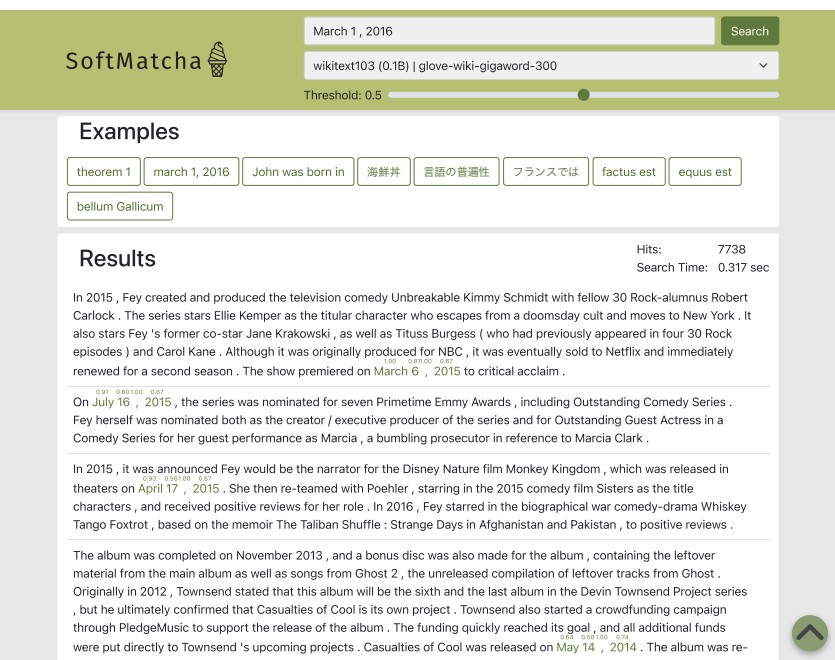

Figure 3: The screenshot of our demo. Given the query "March 1, 2016", it returns lines including softly-matched patterns such as "July 16, 2015."

## D    EFFECTS OF VARYING THRESHOLDS ON SEARCH TIME AND RESULTS

We inspected the relationship between the value of $\alpha$, search time, and the number of matched entries for the query "homemade bomb". We varied the threshold parameter, i.e., $\alpha$ from

---

[7]p.m.

[8]book by Caesar (1882)

[9]Bible, John 13:36

[10]Q.E.D.

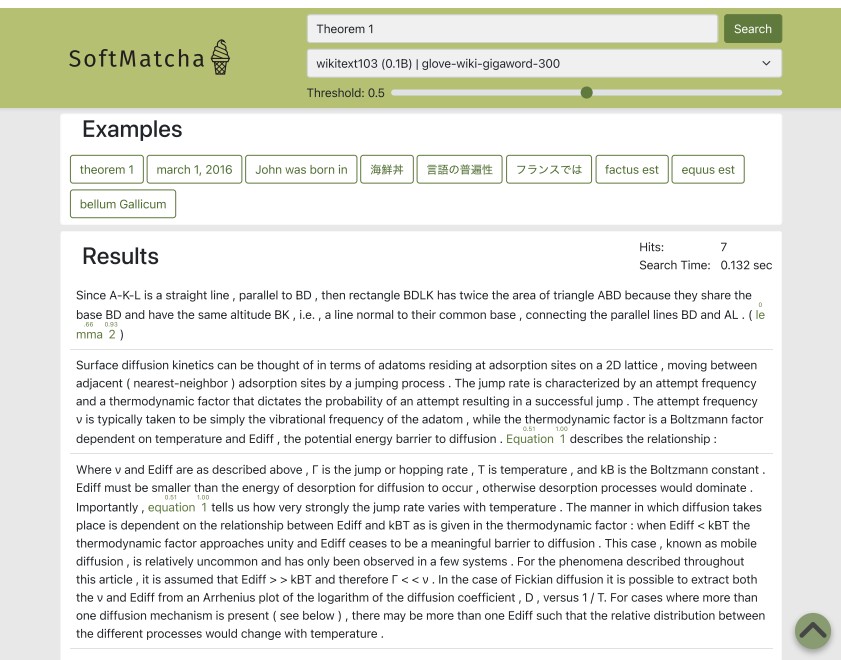

Figure 4: The screenshot of our demo. Given the query "Theorem 1", it returns lines including softly-matched patterns such as "lemma 2."

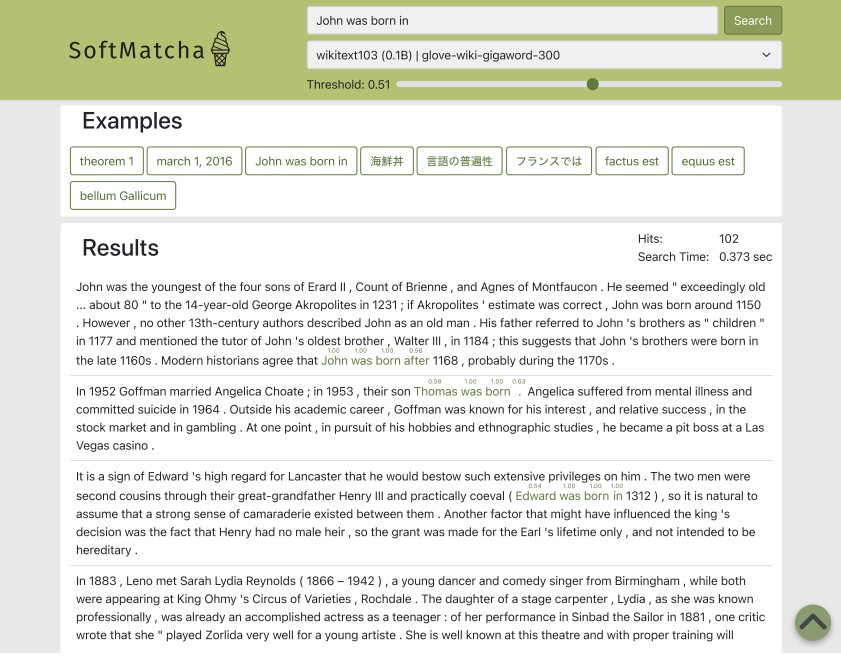

Figure 5: The screenshot of our demo. Given the query "John was born in," it returns lines including softly-matched patterns such as "Edward was born in."

$\{0.1, 0.2, \dots, 1.0\}$, in the English Wikipedia search. The search time and the number of matched entries for each $\alpha$ are shown in Table 7. As you can see, both the search time and the number of matched entries vary significantly depending on the threshold. The results of $\alpha \leq 0.2$ are noteworthy. From the result, we can observe that the search time of the tool is too long to be used for real-time use if the alpha is set below 0.2. However, we remark that the threshold as low as 0.2 is not

Table 7: The search time (seconds) and the number of matched entries for each $\alpha$.

| $\alpha$ | seconds | # of hits |
|---|---|---|
| 0.1 | 1213.503 | 504,955,515 |
| 0.2 | 55.854 | 19,899,652 |
| 0.3 | 1.105 | 183,202 |
| 0.4 | 0.258 | 34,766 |
| 0.5 | 0.079 | 1,839 |
| 0.6 | 0.025 | 381 |
| 0.7 | 0.028 | 259 |
| 0.8 | 0.027 | 259 |
| 0.9 | 0.028 | 107 |
| 1.0 | 0.005 | 107 |

chosen in practice since the search result includes too many nonsensical phrases. For example, the results for $\alpha = 0.2$ included "mixing test" and "authentic scale", which are not related to the query.

# E  ADDITIONAL DISCUSSIONS

## E.1  OUT-OF-VOCABULARY PROBLEM

Our current implementation does not handle out-of-vocabulary (OOV) words; it ignores unknown words. Handling OOV words appropriately is deferred to future work. We plan to (1) use fasttext to fallback unknown words to character embeddings and (2) extend our algorithm to handle subword units to deal with the OOV problem.

## E.2  STATIC EMBEDDINGS VS. DYNAMIC EMBEDDINGS

We used static embeddings for all experiments, but dynamic embeddings could also be used.

The simple one is to use the embedding layer of contextualized models such as BERT (Devlin et al., 2019b), taking an approach like WordLlama (Miller, 2024). We conducted a follow-up experiment using the embedding layer of BERT[11] for the search in the English Wikipedia with a threshold of $\alpha = 0.6$. From the experiment, our SoftMatcha with the embedding layer of BERT newly retrieved "makeshift bomb" and "improvised bomb" etc. with a query of "homemade bombs". It took less than one second, just as fast as when using the GloVe embedding. These new matches may seem a bit counterintuitive, but they suggest that we can possibly obtain deep semantic matches using more dynamic embeddings.

The other more radical one is to actually use contextualized token embeddings instead of mere words for the search keys. Although the contextualized embeddings can be arbitrary real vectors and are not suitable for indexing as is, we can approximate them to a finite number of embeddings by vector quantization or other methods to apply our algorithm. The context information of the query can be strengthened by adding some extra phrases before and after the query.

## E.3  OMISSION, INSERTION, AND SWAPPING

The current SoftMatcha does not treat word omission and insertion, e.g., "the jazz musician" does not match "a fantastic jazz musician". That said, omission and insertion could be handled by modifying Step 2-2 (technically, how to take intersections in line 9, Algorithm 1) using wildcards. Wildcards can skip matching at any position, and is easy to implement by just adjusting the shift width of Step 2-2 in Algorithm 1. This extension allows us to represent various similar patterns with a single simple query.

In addition, it is possible to extend the algorithm to allow the flexible order of words in the query by considering every permutation of the query pattern. This can be quite efficient, because only Step

---

[11]`google-bert/bert-base-uncased`

Table 8: The results of the information retrieval task on the TREC-COVID dataset.

| Method | P@20 | R@1000 | NDCG@20 |
|---|---|---|---|
| BM25 | 39.5 | 22.2 | 34.6 |
| Soft-BM25 ($\alpha = 0.55$) | **41.5** | **23.5** | **36.3** |

2-2 (Find the soft matches) needs to be repeated and typically the pattern length is not too large (technically, dynamic programming can be used here to achieve even better performance).

## F EFFECTIVENESS OF SOFTMATCHA IN THE INFORMATION TASK

We conducted experiments to apply our method to information retrieval.

**Soft-BM25**   The well-known and strong baseline, BM25 (Robertson & Zaragoza, 2009), calculates relevance scores between a query and a document using term frequency (TF) and inverse document frequency (IDF), which involve counting occurrences of words or n-grams.

We implemented a soft version of BM25 (soft-BM25). For calculating the soft term frequency in a document, we compute the pattern-level score by multiplying the matching scores of each word in a query pattern, and then sum up for each pattern occurrence. Similarly, soft inverse document frequency is calculated by counting documents that softly contain the given query patterns. Then, we calculated the soft-BM25 score based on the most standard Lucene implementation.

To examine the effectiveness of soft matching on the information retrieval task, we compared the threshold parameter $\alpha$ between $1.0$ and $0.55$. $\alpha = 1.0$ means "not performing any semantic relaxation", making it equivalent to standard BM25. In contrast, $\alpha = 0.55$ allows for semantic relaxation of query pattern counting using our method, serving as a means to verify the effectiveness of the proposed method in information retrieval.

**Benchmark dataset**   We used TREC-COVID dataset (Roberts et al., 2021), which is a part of Massive Text Embedding Benchmark (MTEB) (Muennighoff et al., 2023). This dataset consists of 171k documents and 50 queries (= instances). Notably, the original dataset comprised only queries formatted as natural language questions that are not suitable for queries of BM25 and soft-BM25; thus, we prepared the pattern queries for this experiment. Specifically, we manually transformed the natural language question, e.g., "how does the coronavirus respond to changes in the weather" to the queries for BM25 and soft-BM25, e.g., ["coronavirus", "response to weather", "weather change", "coronavirus response to weather changes"].

**Evaluation metrics**   We evaluated the retrieval performance on the precision@20 (P@20), recall@1000 (R@1000), NDCG@20 following the previous work (Bendersky et al., 2020).

**Results**   The experimental results are demonstrated in Table 8. The table shows that soft matching achieved better performance compared with exact matching in the information retrieval task.

To summarize this experiment, we confirmed that our SoftMatcha is effective not only in full-text search but also in the information retrieval task, which ranks relevant texts.

