# OpenReview forum: "SoftMatcha: A Soft and Fast Pattern Matcher for Billion-Scale Corpus Searches"
_ICLR.cc/2025/Conference — ICLR 2025 Poster_

### Official Review · Reviewer_4JZq · 2024-11-01

**Soundness:** 4
**Presentation:** 3
**Contribution:** 2
**Rating:** 6
**Confidence:** 5

**Summary:**

This paper introduces a pattern-matching algorithm designed to efficiently handle semantic matching across vast text corpora, addressing the limitations of existing methods that struggle with language variations like orthographic diversity and paraphrasing. Unlike conventional string-based approaches, which lack flexibility, the proposed algorithm leverages word embeddings and inverted indexing to perform "soft" matching, enabling it to find meaningfully related phrases without relying on exact string matches. The algorithm operates at a speed comparable to traditional methods, achieving search times of under a second on billion-word corpora. It proves effective in practical applications: identifying harmful content in large English and Japanese datasets, and retrieving linguistically relevant examples in Latin, where morphological variations are common. SoftMatcha, the resulting tool, is accessible via a web interface, making it practical for users in NLP and digital humanities.

**Strengths:**

The algorithm is optimized for billion-scale corpora, achieving search speeds that are comparable to traditional tools but with a much richer, semantic matching capability. This scalability and speed are significant achievements, especially for applications involving extensive datasets like web-scale corpora.

The paper is well explained and the demo is functional and can be accessed easily. The examples in different languages give a good overview of the applicability potential of the algorithm.

**Weaknesses:**

An important limitation is the reliance on static embeddings, such as GloVe, which have largely been superseded by contextual embeddings in modern NLP applications. The continued use of GloVe and similar embeddings raises questions about the model's ability to capture context-dependent semantics, which are crucial in diverse language patterns and semantic matching tasks. The paper's illustrative example, where semantically "opposite" concepts (e.g., "lived" vs. "died") are matched, does not illustrate, in my opinion, the advantage of using this soft matching but instead it highlights a potential issue, which derives from the use of "classic" embeddings. "live" and "die" are used in similar contexts, which yields a high similarity according to the embedding spaces of word2vec or GloVe, but they are opposite semantically.

I also think that while the paper contains notable technical contributions, the primary focus on linguistic applications, corpus search, and retrieval, with an emphasis on efficiency, suggests that it may align more closely with the interests of venues like ACL rather than ICLR.

**Questions:**

Since the algorithm operates solely on the vocabulary present in the corpus, there is limited discussion on its approach to out-of-vocabulary (OOV) items. This omission raises questions about how the system performs when faced with rare or unseen words and could impact its scalability across languages and domains with rich vocabularies or evolving terminologies.

So my question is: how the matching algorithm deals with OOV words? For instance, if "John" is not in the vocabulary, does it means I can't search for phrases containing "John"?

---

> ### Author Response · Authors · 2024-11-22
> **Reponse to Reviewer 4JZq**
>
> We appreciate your detailed comments and suggestions.  We will consider them carefully and update our revised version of the paper.  Let us address your concerns in the following.
>
> ----
>
> ## Static embeddings vs. Dynamic embeddings
>
> We appreciate your valuable comments. Inspired by your comment, we discovered that we can also apply our algorithm to contextualized embeddings with a slight modification described as follows, reported with our preliminary experimental results.
> The detailed and thorough experimental results will be included in the revised version.
>
> The simple one is to use the embedding layer of contextualized models such as BERT, taking an approach like WordLlama ( https://github.com/dleemiller/WordLlama ).
> We conducted a follow-up experiment using the embedding layer of BERT (bert-base-uncased) for the search in the English Wikipedia (threshold = 0.6). From the experiment, our SoftMatcha with the embedding layer of BERT newly retrieved “makeshift bomb” and “improvised bomb” etc. with a query of “homemade bombs” (taking less than one second, just as fast as when using the GloVe embedding). These new matches may seem a bit counterintuitive, but they suggest that we can possibly obtain deep semantic matches using more dynamic embeddings.
>
> The other more radical one is to actually use contextualized token embeddings instead of mere words for the search keys. Although the contextualized embeddings can be arbitrary real vectors and are not suitable for indexing as is, we can approximate them to a finite number of embeddings by vector quantization or other methods to apply our algorithm. The context information of the query can be strengthened by adding some extra phrases before and after the query.
>
> We would like to emphasize that our proposed algorithm is designed to work with any preferred embeddings, whether static or dynamic.
>
> ----
>
> ## “opposite” concepts
> As you note, whether to include antonyms in the search result depends on the use case.  Here, we argue that including them is indeed important due to two reasons: (1) For corpus linguistics and digital humanities (Bothwell et al. 2023), which is one of the main use cases of SoftMatcha, it is useful to capture such opposite concepts in their search in use cases like parallelism detection; and (2) antonyms are indeed semantically similar to the query word in the sense that they differ in only one semantic feature: polarity.  As an example of (1),  an excerpt from Martin Luther King Jr.’s speech “I have a dream that one day every valley shall be exalted, every hill and mountain shall be made low…” contains a rhetoric parallelism with phrases of opposite semantics.  The present feature of capturing phrases of opposite concepts is useful in detecting such parallelism.
>
> ----
>
> ## Regarding the venue
>
> Thank you for your suggestion.  Although our submission is related to NLP, we believe that it aligns well with the topic “representation learning for computer vision, audio, language, and other modalities” exemplified in the CfP of ICLR 2025 (https://iclr.cc/Conferences/2025/CallForPapers).
>
> ----
>
> ## Out-of-vocabulary problem
>
> > how the matching algorithm deals with OOV words? For instance, if "John" is not in the vocabulary, does it means I can't search for phrases containing "John"?
>
> Yes, our current implementation cannot handle OOV words; it ignores unknown words.  Handling OOV words appropriately is deferred to future work.  We plan to (1) use fasttext to fallback unknown words to character embeddings and (2) extend our algorithm to handle subword units to deal with the OOV problem.  We expect that these extensions are smooth since our algorithm does not assume any particular tokenizer or embedding.

---

> > ### Comment · Reviewer_4JZq · 2024-11-24
> >
> > Thanks to the authors for the explanations. Given this reply I'm leaning towards raising my score.

---

### Official Review · Reviewer_Y3Li · 2024-11-04

**Soundness:** 3
**Presentation:** 3
**Contribution:** 3
**Rating:** 6
**Confidence:** 3

**Summary:**

This paper presents an efficient method for pattern matching over billion-token corpus by combining soft matching via word embeddings and inverted indexes. The method is compared to exact matching and dense vector search as baselines over wikipedia corpora in English and Japanese. They show that the running time of the proposed algorithm is on par with the exact matching but much faster than dense vector search. Through qualitative case analysis, they also show that the algorithm returns relevant results and can be more robust than the other baselines.

**Strengths:**

The algorithm proposed is straightforward and effective, and it can be quite helpful for quickly locating relevant texts in massive pretraining data. The paper is also well-written and structured clearly, especially the demo interface.

**Weaknesses:**

The empirical evaluation can be further strengthened. It would be helpful to include analysis that shows 1) how the threshold influences speed and the relevance of the retrieved results. 2) a tradeoff analysis between the efficiency and the qualitative analysis. Table 3 seems to show that SoftMatcha retrieve relevant and robust texts fast. However I wonder if the observed relevancy can be quantified with a information retrieval corpora so that we can understand its relevant performance compared to the baselines.

**Questions:**

n/a

---

> ### Author Response · Authors · 2024-11-22
> **Response to Reviewer Y3Li**
>
> We appreciate your taking the time to review this paper. We will answer your questions and comments, and also update our revised manuscript. Let us address your concerns in the following.
>
> ----
>
> ## Additional empirical evaluation
> > The empirical evaluation can be further strengthened. It would be helpful to include analysis that shows 1) how the threshold influences speed and the relevance of the retrieved results. 2) a tradeoff analysis between the efficiency and the qualitative analysis.
>
> Thank you for your question.  This question is very important since its answer directly affects the user experience of the tool.  To answer your question, we quickly examined the relationship between the value of $\alpha$, search time, and the number of matched entries for the query “homemade bomb”, as shown in Table 3.  We varied the threshold parameter, i.e., $\alpha$ from $\lbrace 0.1, 0.2, …, 1.0 \rbrace$, in the English Wikipedia search. The search time and the number of matched entries for each alpha were as follows:
>
> | $\alpha$ | search time (seconds) | # matched entries |
> |----|----|----|
> | 0.1 | 1213.503 | 504,955,515 |
> | 0.2 | 55.854 | 19,899,652 |
> | 0.3 | 1.105 | 183,202 |
> | 0.4 | 0.258 | 34,766 |
> | 0.5 | 0.079 | 1,839 |
> | 0.6 | 0.025 | 381 |
> | 0.7 | 0.028 | 259 |
> | 0.8 | 0.027 | 259 |
> | 0.9 | 0.028 | 107 |
> | 1.0 | 0.005 | 107 |
>
> As you can see, both the search time and the number of matched entries vary significantly depending on the threshold $\alpha$. This comparison which you suggested certainly seems valuable. We would like to include the results of these experiments in our revised manuscript.
>
> The results of $\alpha \leq 0.2$ are noteworthy.
> From the result, we can observe that the search time of the tool is too long to be used for real-time use if alpha is set below 0.2.  However, we remark that the threshold as low as 0.2 is not chosen in practice since the search result includes too many nonsensical phrases.  For example, the results for $\alpha=0.2$ included “mixing test” and “authentic scale”, which are not related to the query.  We may need to inspect the relationship between alpha and relevance of the search results and change the web interface so that one can choose alpha only from the range that produces the relevant results.  We will include these discussions in the updated version.
>
> ----
>
> ## Regarding information retrieval and ranking the search results
> > Table 3 seems to show that SoftMatcha retrieve relevant and robust texts fast. However I wonder if the observed relevancy can be quantified with a information retrieval corpora so that we can understand its relevant performance compared to the baselines.
>
> Thank you for the valuable comment.  It is indeed natural to recall ranking problems such as information retrieval from our setting of matching documents from a query. Other reviewers also asked about an evaluation of the information retrieval task and I understand that this is an important point to improve the clarity of our manuscript.  Let us address your concern below, which will be incorporated in the revised version.
>
> We first would like to remark that our method is intended for *enumerating* all matched spans, like grep in CLI environment and concordancers in corpus linguistics.  However, the information retrieval tools such as Elasticsearch are intended to be used for *ranking*.  Due to this difference, the format of the output of these two is different from each other.
>
> That said, an extension of our method for ranking is an attractive direction.  We have the following ideas for this extension.
> One idea is to rank entries by the match score itself, which uses the pooled match score over the pattern. A possible problem with this is that many matches can have the same rank if the same set of words appears at many positions in the text.
> Another idea is to rerank by other rank-based retrievers, such as BM25 or dense vector search, combiging several search methods depending on the purpose.

---

> > ### Comment · Reviewer_Y3Li · 2024-11-26
> >
> > I would like to thank the authors for their clarification.
> >
> > Regarding comparison with IR tasks, I understand the objective could be different. However since IR methods could capture relevance signals from ngram matching or word frequencies, it would be a good baseline to see how many exact spans they are able to capture compared to your method.

---

> ### Author Response · Authors · 2024-12-04
> **Response to Reviewer Y3Li**
>
> ## Experiments of the information retrieval task
>
> Thank you for your insightful comments.
> From your comments “IR methods could capture relevance signals from n-gram matching or word frequencies”, we noticed that we could calculate the *soft* term frequency and *soft* inversed document frequency by softly counting the frequencies of query patterns with our method, and thus could realize the soft version of BM25 (*soft-BM25*).
>
> We promptly conducted experiments to apply our method to information retrieval.
> Encouragingly, the experimental results demonstrate that, by relaxing this discrete counting of words or word-level n-grams with our SoftMatcha, the search accuracy of BM25 has been improved. Specifically, both recall and precision were enhanced through this approach.
>
> We will include these experimental results in the revised manuscript.
>
> ### Soft-BM25
> We implemented a soft version of BM25 (soft-BM25).
> For calculating the *soft* term frequency in a document, we compute the pattern-level score by multiplying the matching scores of each word in a query pattern, and then sum up for each pattern occurrence.
> Similarly, *soft* inverse document frequency is calculated by counting documents that softly contain the given query patterns.
> Then, we calculated the soft-BM25 score based on the most standard Lucene implementation.
>
> To examine the effectiveness of soft matching on the information retrieval task, we compared the threshold parameter $\alpha$ between $1.0$ and $0.55$.
> $\alpha=1.0$ means “not performing any semantic relaxation”, making it equivalent to standard BM25. In contrast, $\alpha=0.55$ allows for semantic relaxation of query pattern counting using our method, serving as a means to verify the effectiveness of the proposed method in information retrieval. For this experiment, we did not tune the $\alpha$ and set it to the same value as that employed in the current manuscript.
>
> ### Benchmark dataset
> We used [trec-covid dataset](https://huggingface.co/datasets/mteb/trec-covid) for the experiment, which is included in [MTEB: Massive Text Embedding Benchmark](https://arxiv.org/abs/2210.07316). This dataset consists of 171k documents and 50 queries (= instances).
> Notably, the original dataset comprised only queries formatted as natural language questions that are not suitable for queries of BM25 and soft-BM25. Thus, we prepared the pattern queries for this experiment. For example:
>     - Natural language question: "how does the coronavirus respond to changes in the weather"
>     - Queries for BM25 and soft-BM25: ["coronavirus", "response to weather", “weather change”, “coronavirus response to weather changes”]
>
> ### Evaluation metrics
> The retrieval accuracy is evaluated on the precision@20 (P@20), recall@1000, NDCG@20 following [the previous work](https://arxiv.org/abs/2010.00200).
>
>  ### Results
>
> | Method | P@20 | recall@1000 | NDCG@20 |
> | ---- | ---- | ---- | ---- |
> | BM25 | 39.5 | 22.2 | 34.6 |
> | Soft-BM25 ($\alpha$ = 0.55) | **41.5** | **23.5** | **36.3** |
>
> The table shows that soft matching achieved better performance compared with exact matching in the information retrieval task.
>
> To summarize this experiment, we confirmed that our SoftMatcha is effective not only in full-text search but also in the information retrieval task, which ranks relevant texts.

---

### Official Review · Reviewer_9DuR · 2024-11-04

**Soundness:** 2
**Presentation:** 4
**Contribution:** 3
**Rating:** 5
**Confidence:** 3

**Summary:**

Paper addresses the problem of soft (or semantic) pattern matching over a billion-scale text corpora. Exact matching is often too stringent; on the other hand, naive semantic matching based on computing similarities between word embeddings is much slower, since it requires a lot of floating point operations with high-dimensional vectors instead of fast bitwise comparisons.

Authors leverage inverted indexes data structure to reduce the amount of required soft matches, and propose an algorithm which requires only a constant amount of soft matches with respect to corpura size. They also conduct two case studies: testing the speed of the proposed implementation on a billion-scale search, and retrieving patterns from a morphologically complex language (Latin). Finally, they provide a demo to interact with the proposed algorithm.

**Strengths:**

- Paper is very clearly written and easy to follow.

- Main algorithm is simple and straightforward Notably, it requires a constant amount of costly soft word comparisons (which can be ~1000 longer compared to hard comparisons) with respect to corpus size Meanwhile, it still allows for much more flexible matching.

- Provided demo works.

**Weaknesses:**

1. Description of dense vector search lacks details.
- e.g. in Line 415, what is a “training instance” — a sentence? A paragraph?
- How the training instances and queries were formatted for the model? Does it follow the description at https://huggingface.co/intfloat/multilingual-e5-large?
- What was the maximal length of training instances encoded by the model?
If maximal length of a training instance is too long for the model, was it truncated or split into chunks?
- How token embeddings were pooled in a single chunks-level score?
- What were the HNSW hyperparameters?

2. Table 2 lacks details.
- How many search queries were done (e.g. 1, 100, 1000)?
- How long the queries were (e.g. <20 words)?
- What are the variation in search time (standard deviation of search time over multiple queries, maximal / minimal search time for a single query)?

3. Approximate string matching baselines were not included (e.g. agrep (Wu & Manber, 1992b;a) or TRE (https://en.wikipedia.org/wiki/TRE_(computing))). While unlike SoftMatcha they can not e.g. find synonyms, they are still useful for finding matches with typos / inflected word forms (which e.g. was part of the goal in “Section 4.3 Case study in computational linguistics — retrieving Latin examples), and it is interesting to compare search speed.
- In Section 2.3 String Matching, lines 200-202 you state that approximate string matching is orthogonal to your work as it focuses on surface-level comparison, while you target semantic-level similarity.
However, in Introduction, lines 076-080 you mention that “...it is often desirable to catch non-standard spellings as well, such as how r u instead of how are you … Additionally, it is desirable to catch different inflected word forms such as sing, sang, sung, signs and singing, which differ only in their morphological features and share the same lemma (base form).”

One big difficulty with assigning an overall score is the fact, that there are basically no metrics in the paper, except for running time.
For approximate/fuzzy pattern matching algorithms it is interesting to estimate how relevant are the search results, using ranking metrics like mean average precision, or NDCG. However, these require a labeled dataset, which is hard to obtain.

On a qualitative level, this algorithm looks like a noticeable improvement over exact matching, and it might have some benefits compared to dense vector search (e.g. 100% recall in cases where the query exactly matches the pattern, higher speed, simpler implementation). Nonetheless, with more search results there arises a need to rank them. Considering Figure 3: while in some cases the ability to match “July 16, 2015” by the query “March 1, 2016” is useful, in others this might be a completely irrelevant search result.

**Questions:**

Major:
- There are some questions in Weaknesses section.
- How the sample search results were selected for Table 3?

Minor:
- How to choose alpha?
- How much more search results does SoftMatcha retrieve depending on alpha?
    - How many of additional entries are irrelevant?
    - How to rank additional entries found by Softmatcha?
- For potential future work: is it possible to modify the algorithm to allow flexible order of words in the query, or matching with some words omitted / inserted (e.g. finding “a fantastic jazz musician” by a query “the jazz musician”)? Fixed word order is still quite restrictive, especially for queries containing many words.

Suggestions:

- In footnote 5, page 8 information about Japanese as fifth language in Wikipedia by total amount of articles is outdated — Japanese is now 17th, while Chinese is 10th with almost twice as many total articles. Source: https://wikistats.wmcloud.org/display.php?t=wp. I propose to update the footnote.
- Lines 099-100: “0.1 seconds … without GPU” — I’d propose to add hardware specifications used for speed evaluations (156 cores and 226 Gib of main memory), to make the claim more concrete.

---

> ### Author Response · Authors · 2024-11-22
> **Response to Reviewer 9DuR**
>
> We appreciate your detailed comments and suggestions. We will consider them carefully and reflect them in our revised version.  The following is the answer to your comments.
>
> ----
>
> ## Detailed descriptions of dense vector search
>
> > e.g. in Line 415, what is a “training instance” — a sentence? A paragraph?
>
> “Training instance” there refers to a chunk of text, which could be a single sentence or multiple sentences. It means the mini-batches that were actually used in the LLM training,  the chunks to be searched in our experiments.
>
> > How the training instances and queries were formatted for the model?
> > Does it follow the description at https://huggingface.co/intfloat/multilingual-e5-large?
>
> Yes, we followed the description of https://huggingface.co/intfloat/multilingual-e5-large; we pretended the prefix “passage: ” to each chunk of text and “query: ” to each query.
>
> > What was the maximal length of training instances encoded by the model? If maximal length of a training instance is too long for the model, was it truncated or split into chunks?
>
> The maximal length was set to 512 tokens, which corresponds to the maximum length supported by the multilingual-e5-large tokenizer and its positional embeddings. We truncated tokens that exceed 512.
>
> > How token embeddings were pooled in a single chunks-level score?
>
> We averaged the contextualized token embeddings following the description of https://huggingface.co/intfloat/multilingual-e5-large.
>
> > What were the HNSW hyperparameters?
>
> We used the faiss implementation for HNSW with the number of edges set to 32. The `efConstruction` parameter was set to 40, while `efSearch` was configured as 16, which are the default parameters in the faiss implementation.
>
> We will describe these details of settings in the revised manuscript.
>
> ----
>
> ## Details of Table 2
>
> Thank you for pointing it out. We will include the following explanation in the revised version, which will improve the reproducibility of our experiments.
>
> > How many search queries were done (e.g. 1, 100, 1000)?
> > How long the queries were (e.g. <20 words)?
>
> Table 3 shows the running time of a single query “homemade bomb”, consisting of two tokens.
>
> > What are the variation in search time (standard deviation of search time over multiple queries, maximal / minimal search time for a single query)?
>
> We additionally conducted the same search ten times. For exact match search, the average search time was 0.009 seconds with a standard deviation of 0.009 seconds. Meanwhile, in our soft match search (with α = 0.55), the average search time was 0.114 seconds with a standard deviation of 0.021 seconds. Dense vector search took an average time of 0.431 seconds with a standard deviation of 0.053 seconds.
>
> We will add these descriptions to the section of experiments.
>
> ----
>
> ## Approximate string match
>
> This is indeed an important point.
> Let us elaborate on the relationship between approximate string matching and our method below.
>
> Surface-level similarity between two words (defined, for example, based on their edit distance) often works very differently from their semantic similarity; in this sense, we argue that these two similarity notions are orthogonal.  For example, strings like “desert” and “dessert” (Levenshtein distance of 1) may match in approximate string matching, while our method likely ignores them given their semantic irrelevance. On the other hand, morphologically irregular forms such as “person” and “people” (Levenshtein distance of 4), only differing in plurality, will likely be missed in approximate string matching, while our methods are able to catch them.
> Since approximate string matching is conditioned by strings and ours by token embeddings, we can enhance our method via integration with approximate string matching. For example, our present method cannot handle the cases in which adjectives, adverbs, and prepositional phrases are inserted in the middle of a phrase, e.g., “He is actually kind” and “Tom is kind”, which edit-distance-based similarity can handle better. This observation can lead to a promising future direction of integrating our method and word-level Levenshtein distance.  Thank you very much for your inspiring comment. We will add this discussion in the revised version in detail.

---

> ### Author Response · Authors · 2024-11-22
>
> ## Rank of the search results
>
> > One big difficulty with assigning an overall score is the fact, that there are basically no metrics in the paper, except for running time. For approximate/fuzzy pattern matching algorithms it is interesting to estimate how relevant are the search results, using ranking metrics like mean average precision, or NDCG.
>
> Thank you for the valuable comment.  It is indeed natural to recall ranking problems such as information retrieval from our setting of matching documents from a query.  Let us address your concern below, which will be incorporated in the camera-ready version.
>
> We first would like to remark that our method is intended for *enumerating* all matched spans, like grep in CLI environment and concordancers in corpus linguistics.  However, the information retrieval tools such as Elasticsearch are intended to be used for *ranking*.  Due to this difference, the format of the output of these two is different from each other.
>
> That said, an extension of our method for ranking is an attractive direction.  We have the following ideas for this extension.
> One idea is to rank entries by the match score itself, which uses the pooled match score over the pattern. A possible problem with this is that many matches can have the same rank if the same set of words appears at many positions in the text.
> Another idea is to rerank by other rank-based retrievers, such as BM25 or dense vector search, combiging several search methods depending on the purpose.

---

> ### Author Response · Authors · 2024-11-22
>
> ## Questions that you pointed out
>
> > How the sample search results were selected for Table 3?
>
> We picked up several toxic queries that are frequently used to filter corpus for AI safety aiming at deterring terrorism.  The search results in Table 3 are randomly sampled from the entire results due to page limitations.  We recognize the importance of presenting diverse examples to demonstrate the importance of our tool.  In the revised version, we will incorporate more examples in the appendix that are as exhaustive as we can.  For example, we will include an example in which “illicit drug selling” hits for the query “illegal drug sales” in the English Wikipedia search.
>
> > How to choose alpha?
>
> We decided on the threshold alpha subjectively after trying several values and observing the results. We plan to study the automated selection of alpha in future work.
>
> > How much more search results does SoftMatcha retrieve depending on alpha?
> > How many of additional entries are irrelevant?
>
> Thank you for your question.  This question is very important since its answer directly affects the user experience of the tool.  To answer your question, we inspected the relationship between the value of $\alpha$, search time, and the number of matched entries for the query “homemade bomb”.  We varied the threshold parameter, i.e., $\alpha$ from $\lbrace 0.1, 0.2, …, 1.0 \rbrace$, in the English Wikipedia search. The search time and the number of matched entries for each $\alpha$ were as follows:
>
> | $\alpha$ | search time (seconds) | # matched entries |
> |----|----|----|
> | 0.1 | 1213.503 | 504,955,515 |
> | 0.2 | 55.854 | 19,899,652 |
> | 0.3 | 1.105 | 183,202 |
> | 0.4 | 0.258 | 34,766 |
> | 0.5 | 0.079 | 1,839 |
> | 0.6 | 0.025 | 381 |
> | 0.7 | 0.028 | 259 |
> | 0.8 | 0.027 | 259 |
> | 0.9 | 0.028 | 107 |
> | 1.0 | 0.005 | 107 |
>
> As you can see, both the search time and the number of matched entries vary significantly depending on the threshold $\alpha$. This comparison which you suggested certainly seems valuable. We would like to include the results of these experiments in our revised manuscript.
>
> The results of $\alpha \leq 0.2$ are noteworthy.
> From the result, we can observe that the search time of the tool is too long to be used for real-time use if alpha is set below 0.2.  However, we remark that the threshold as low as 0.2 is not chosen in practice since the search result includes too many nonsensical phrases.  For example, the results for $\alpha=0.2$ included “mixing test” and “authentic scale”, which are not related to the query.  We may need to inspect the relationship between alpha and relevance of the search results and change the web interface so that one can choose alpha only from the range that produces the relevant results.  We will include these discussions in the updated version.
>
> > How to rank additional entries found by Softmatcha?
>
> Our SoftMatcha is a kind of full-text search algorithm, so currently it just enumerates all the matches in order without ranking.
> Still, as future topics, we can consider ranking methods, such as ranking by the match score itself or by cascaded BM25 reranker.
>
> > For potential future work: is it possible to modify the algorithm to allow flexible order of words in the query, or matching with some words omitted / inserted (e.g. finding “a fantastic jazz musician” by a query “the jazz musician”)? Fixed word order is still quite restrictive, especially for queries containing many words.
>
> Thank you for suggesting an interesting extension.
> Omission and insertion can be handled by modifying Step 2-2 (technically, how to take intersections in Line 9, Algorithm 1). In particular, we plan to discuss and implement the wildcard pattern in future work or the final submission. Wildcard can skip matching at any position, and is easy to implement by just adjusting the shift width of Step 2-2 in Algorithm 1.
> This extension allows us to represent various similar patterns with a single simple query. Thank you for your attractive suggestion.
>
> Flexible order can also be handled by considering every permutation of the query pattern. This can be quite efficient, because only Step 2-2 (Find the soft matches) needs to be repeated and typically the pattern length is not too large (technically, dynamic programming can be used here to achieve even better performance). We plan to discuss and implement this in future work or the final submission.

---

> ### Author Response · Authors · 2024-11-22
>
> ## Regarding your suggestions
>
> Thank you for your detailed comments to help improve our paper. We will reflect your suggestions in the revised manuscript.
>
> > In footnote 5, page 8 information about Japanese as fifth language in Wikipedia by total amount of articles is outdated — Japanese is now 17th, while Chinese is 10th with almost twice as many total articles. Source: https://wikistats.wmcloud.org/display.php?t=wp. I propose to update the footnote.
>
> We appreciate your valuable information. Upon verifying the source, the source and confirmed that Japanese is now 17th, while Chinese is 10th with almost twice as many total articles. We will update our description with the latest statistics.
>
> > Lines 099-100: “0.1 seconds … without GPU” — I’d propose to add hardware specifications used for speed evaluations (156 cores and 226 Gib of main memory), to make the claim more concrete.
>
> This is also as you pointed out. We will also add the hardware specs to the introduction to clarify our claim.

---

> > ### Comment · Reviewer_9DuR · 2024-11-29
> > **Answer to the Authors**
> >
> > Given clarification about the scope of the paper (specifically, the focus on “enumerating” rather than ranking matches) I am leaning towards keeping my original score.
> >
> > Rationale:
> > Enumerating matches could be a part of search / information retrieval pipeline. In such a case, the ultimate metric would be the relevance of search results. Here I second Reviewer Y3Li that paper would benefit greatly from extending the original approach with some form of reranking (by similarity score, or BM25, as mentioned in Author’s response) and including evaluations on information retrieval benchmarks (e.g. something from Retrieval in MTEB [1]).
> > Enumerating matches could be considered in isolation, leaving the question of reranking out of the scope, but then I expect smaller impact from this paper as is.
> > In the Introduction, lines 048-070 Authors mention following possible applications:
> > finding training instances responsible for harmful information, misinformation or private information;
> > studying rare linguistic phenomenons.
> > Authors provide a qualitative case study for b), but this looks as a quite specific and narrow application to me. Adding more case studies where an “match enumerator” without a reranking procedure shows promising results would be an alternative way to strengthen the paper.
> >
> > I want to acknowledge the Authors’ effort in providing detailed responses about dense vector search, Tables 2 and 3, and thank for the additional experiments regarding sensitivity to `alpha`, extending Table 3, adding information about hardware specification and updating footnote about Wikipedia articles statistics.
> > Still, for Table 2 I’d propose to measure speed over a more comprehensive set of queries (at least 100), ideally additionally controlling for the length of query (I suppose, it might significantly impact the search time).
> >
> > [1] MTEB: Massive Text Embedding Benchmark, https://arxiv.org/abs/2210.07316

---

> ### Author Response · Authors · 2024-12-04
> **Response to Reviewer 9DuR**
>
> ## Experiments of the information retrieval task
> Thank you for your additional explanations!
> We have finally understood the importance of evaluating our proposed method in the context of information retrieval task as well.
>
> We promptly conducted experiments to apply our method to information retrieval.
> The well-known and strong baseline, BM25, calculates relevance scores between a query and a document using term frequency (TF) and inverse document frequency (IDF), which involve counting occurrences of words or n-grams.
> Encouragingly, the experimental results demonstrate that, by relaxing this discrete word or n-gram counts with our SoftMatcha, the search accuracy of BM25 has been improved. Specifically, both recall and precision were enhanced through this approach.
>
> Again, we appreciate your valuable suggestion.
> We will include more comprehensive experimental results in the revised manuscript.
>
> ### Soft-BM25
> We implemented a soft version of BM25 (soft-BM25).
> For calculating the *soft* term frequency in a document, we compute the pattern-level score by multiplying the matching scores of each word in a query pattern, and then sum up for each pattern occurrence.
> Similarly, *soft* inverse document frequency is calculated by counting documents that softly contain the given query patterns.
> Then, we calculated the soft-BM25 score based on the most standard Lucene implementation.
>
> To examine the effectiveness of soft matching on the information retrieval task, we compared the threshold parameter $\alpha$ between $1.0$ and $0.55$.
> $\alpha=1.0$ means “not performing any semantic relaxation”, making it equivalent to standard BM25. In contrast, $\alpha=0.55$ allows for semantic relaxation of query pattern counting using our method, serving as a means to verify the effectiveness of the proposed method in information retrieval. For this experiment, we did not tune the $\alpha$ and set it to the same value as that employed in the current manuscript.
>
> ### Benchmark dataset
> We used [trec-covid dataset](https://huggingface.co/datasets/mteb/trec-covid) for the experiment, which is included in [MTEB: Massive Text Embedding Benchmark](https://arxiv.org/abs/2210.07316). This dataset consists of 171k documents and 50 queries (= instances).
> Notably, the original dataset comprised only queries formatted as natural language questions that are not suitable for queries of BM25 and soft-BM25. Thus, we prepared the pattern queries for this experiment. For example:
>     - Natural language question: "how does the coronavirus respond to changes in the weather"
>     - Queries for BM25 and soft-BM25: ["coronavirus", "response to weather", “weather change”, “coronavirus response to weather changes”]
>
> ### Evaluation metrics
> The retrieval accuracy is evaluated on the precision@20 (P@20), recall@1000, NDCG@20 following [the previous work](https://arxiv.org/abs/2010.00200).
>
>  ### Results
>
> | Method | P@20 | recall@1000 | NDCG@20 |
> | ---- | ---- | ---- | ---- |
> | BM25 | 39.5 | 22.2 | 34.6 |
> | Soft-BM25 ($\alpha$ = 0.55) | **41.5** | **23.5** | **36.3** |
>
> The table shows that soft matching achieved better performance compared with exact matching in the information retrieval task.
>
> To summarize this experiment, we confirmed that our SoftMatcha is effective not only in full-text search but also in the information retrieval task, which ranks relevant texts.
>
> ----
>
> ## Factors that affect search speed
> Thank you for your valuable feedback on the performance of our algorithm for various patterns. We would like to clarify that the time complexity of soft matching is, at most, linear in the pattern length. This implies that the execution time is unproblematic in practice since the pattern length is typically at most ~20 words. More precisely, as we explain below, the computational cost of our algorithm also depends on the number of occurrences of the words that softly match the pattern words. In our final version, we will add this theoretical complexity analysis and more extensive experiments with diverse queries to clarify this aspect.
> Below is a sketch of the complexity analysis. As explained in Section 3.3, the number of soft membership evaluations (i.e., line 4 of Algorithm 1) is exactly $n \times L$, where $n$ is the pattern length and $L$ is the vocabulary size. Therefore, it is linear in the pattern length. The number of the union operations (i.e., line 7 of Algorithm 1) is $\sum_{k \in \{1,\dots,n\}} |S_k|$, where $S_k$ is the set of words that softly match the $k$-th word in the pattern. Therefore, it is also linear in the pattern length. More precisely, the computational cost also depends on how many times the words in $S_k$ occur in the corpus. The number of intersection operations (i.e., line 9 of Algorithm 1) is also linear in the pattern length. Again, more precisely, the computational cost also depends on the number of the current match candidates $|M|$ and how many times the words in $S_k$ occur in the corpus.

---

### Comment · Area_Chair_qpqG · 2024-11-21
**Reminder: Please respond and update the score if necessary**

Dear Reviewers,

Kindly ensure that you respond proactively to the authors' replies (once they are available) so we can foster a productive discussion. If necessary, please update your score accordingly. We greatly appreciate the time and effort you’ve dedicated to the review process, and your contributions are key to making this process run smoothly.

Thank you,

AC

---

> ### Author Response · Authors · 2024-11-24
>
> Thank you for your reminder to encourage discussion.
> We have submitted author responses to each reviewer.
>
> We are pleased to have received many constructive comments and hope to continue these fruitful discussions.

---

### Meta-Review · Area_Chair_qpqG · 2024-12-22

**Metareview:**

This paper presents an algorithm for semantic pattern matching across large-scale text corpora, addressing the shortcomings of exact and traditional semantic matching methods. By combining word embeddings with inverted indexing, the proposed method achieves efficient "soft" matching. It maintains search speeds comparable to exact matching while being faster and more robust than dense vector search. Tested on English, Japanese, and Latin corpora, the algorithm effectively balances speed and semantic relevance. The resulting tool is accessible via a web interface, making it useful for various applications in natural language processing and digital humanities.

On the positive note, reviewers agree that the paper introduces a straightforward and effective algorithm optimized for billion-scale corpora, offering search speeds comparable to traditional tools while providing enhanced semantic matching capabilities. This significant advancement is particularly useful for applications involving extensive datasets, such as web-scale corpora. The paper is well-written and structured clearly, making it easy to follow. The demonstration interface is functional and accessible, with examples in various languages showcasing the algorithm's broad applicability. Notably, the algorithm requires a constant amount of computationally intensive soft word comparisons relative to corpus size, ensuring flexible matching without compromising speed.

The reviewers have highlighted key areas needing enhancement, particularly the necessity for a more detailed empirical evaluation and analysis of how threshold settings impact speed and result relevance. The paper's reliance on static embeddings such as GloVe is noted as a limitation, given that these are less effective than contemporary contextual embeddings at capturing nuanced semantics. Additionally, the reviewers remarked that, despite the paper's notable technical contributions, its primary focus on linguistic applications and efficiency may not align well with venues centered on representation learning.

Overall, the paper has significant merits, I recommend acceptance.

**Additional Comments On Reviewer Discussion:**

Reviewers identified several areas for improvement in the paper. Reviewer 9DuR noted a lack of detail in the description of dense vector search. Reviewer Y3Li suggested enhancing the empirical evaluation by analyzing the impact of thresholds on speed and result relevance and conducting a trade-off analysis between efficiency and qualitative outcomes. They also recommended using an information retrieval corpus to better quantify the relevance of SoftMatcha's results. Reviewer 4JZq highlighted a significant limitation: the reliance on static embeddings like GloVe, which are not as effective as modern contextual embeddings in capturing context-dependent semantics. Additionally, while the paper makes notable technical contributions, its primary focus on linguistic applications and efficiency may align more closely with venues like ACL rather than ICLR.

---

### Decision · Program_Chairs · 2025-01-22

Accept (Poster)